# Sub-sampled Newton Methods
# with Non-uniform Sampling

**Peng Xu**[†] **Jiyan Yang**[†] **Farbod Roosta-Khorasani**[‡] **Christopher Ré**[†] **Michael W. Mahoney**[‡]
† Stanford University ‡ University of California at Berkeley
pengxu@stanford.edu jiyan@stanford.edu farbod@icsi.berkeley.edu
chrismre@cs.stanford.edu mmahoney@stat.berkeley.edu

## Abstract

We consider the problem of finding the minimizer of a convex function $F : \mathbb{R}^d \to \mathbb{R}$ of the form $F(\mathbf{w}) := \sum_{i=1}^{n} f_i(\mathbf{w}) + R(\mathbf{w})$ where a low-rank factorization of $\nabla^2 f_i(\mathbf{w})$ is readily available. We consider the regime where $n \gg d$. We propose randomized Newton-type algorithms that exploit *non-uniform* sub-sampling of $\{\nabla^2 f_i(\mathbf{w})\}_{i=1}^{n}$, as well as inexact updates, as means to reduce the computational complexity, and are applicable to a wide range of problems in machine learning. Two non-uniform sampling distributions based on *block norm squares* and *block partial leverage scores* are considered. Under certain assumptions, we show that our algorithms inherit a linear-quadratic convergence rate in $\mathbf{w}$ and achieve a lower computational complexity compared to similar existing methods. In addition, we show that our algorithms exhibit more robustness and better dependence on problem specific quantities, such as the condition number. We empirically demonstrate that our methods are at least twice as fast as Newton's methods on several real datasets.

## 1 Introduction

Many machine learning applications involve finding the minimizer of optimization problems of the form

$$\min_{\mathbf{w} \in \mathcal{C}} F(\mathbf{w}) := \sum_{i=1}^{n} f_i(\mathbf{w}) + R(\mathbf{w}) \tag{1}$$

where $f_i(\mathbf{w})$ is a smooth convex function, $R(\mathbf{w})$ is a regularizer, and $\mathcal{C} \subseteq \mathbb{R}^d$ is a convex constraint set (e.g., $\ell_1$ ball). Examples include sparse least squares [21], generalized linear models (GLMs) [8], and metric learning problems [12].

First-order optimization algorithms have been the workhorse of machine learning applications and there is a plethora of such methods [3, 17] for solving (1). However, for ill-conditioned problems, it is often the case that first-order methods return a solution far from $\mathbf{w}^*$ albeit a low objective value. On the other hand, most second-order algorithms prove to be more robust to such adversarial effects. This is so since, using the curvature information, second order methods properly rescale the gradient, such that it is a more appropriate direction to follow. For example, take the canonical second order method, i.e., *Newton's method*, which, in the unconstrained case, has updates of the form $\mathbf{w}_{t+1} = \mathbf{w}_t - [\mathbf{H}(\mathbf{w}_t)]^{-1}\mathbf{g}(\mathbf{w}_t)$ (here, $\mathbf{g}(\mathbf{w}_t)$ and $\mathbf{H}(\mathbf{w}_t)$ denote the gradient and the Hessian of $F$ at $\mathbf{w}_t$, respectively). Classical results indicate that under certain assumptions, Newton's method can achieve a locally super-linear convergence rate, which can be shown to be *problem independent*! Nevertheless, the cost of forming and inverting the Hessian is a major drawback in using Newton's method in practice. In this regard, there has been a long line of work aiming at providing sufficient second-order information more efficiently, e.g., the classical BFGS algorithm and its limited memory version [14, 17].

As the mere evaluation of $\mathbf{H}(\mathbf{w})$ grows linearly in $n$, a natural idea is to use uniform sub-sampling $\{\nabla^2 f_i(\mathbf{w})\}_{i=1}^{n}$ as a way to reduce the cost of such evaluation [7, 19, 20]. However, in the presence of high non-uniformity among $\{\nabla^2 f_i(\mathbf{w})\}_{i=1}^{n}$, the sampling size required to sufficiently capture the

curvature information of the Hessian can be very large. In such situations, *non-uniform* sampling can indeed be a much better alternative and is addressed in this work in detail.

In this work, we propose novel, robust and highly efficient non-uniformly sub-sampled Newton methods (SSN) for a large sub-class of problem (1), where the Hessian of $F(\mathbf{w})$ in (1) can be written as $\mathbf{H}(\mathbf{w}) = \sum_{i=1}^{n} \mathbf{A}_i^T(\mathbf{w})\mathbf{A}_i(\mathbf{w}) + \mathbf{Q}(\mathbf{w})$, where $\mathbf{A}_i(\mathbf{w}) \in \mathbb{R}^{k_i \times d}$, $i = 1, 2, \ldots, n$, are readily available and $\mathbf{Q}(\mathbf{w})$ is some positive semi-definite matrix. This situation arises very frequently in machine learning problems. For example, take any problem where $f_i(\mathbf{w}) = \ell(\mathbf{x}_i^T \mathbf{w})$, $\ell(\cdot)$ is any convex loss function and $\mathbf{x}_i$'s are data points. In such situations, $\mathbf{A}_i(\mathbf{w})$ is simply $\sqrt{\ell''(\mathbf{x}_i^T \mathbf{w})}\mathbf{x}_i^T$. Under this setting, non-uniformly sub-sampling the Hessians now boils down to building an appropriate non-uniform distribution to sub-sample the most "relevant" terms among $\{\mathbf{A}_i(\mathbf{w})\}_{i=1}^{n}$. The approximate Hessian, denoted by $\widetilde{\mathbf{H}}(\mathbf{w}_t)$, is then used to update the current iterate as $\mathbf{w}_{t+1} = \mathbf{w}_t - [\widetilde{\mathbf{H}}(\mathbf{w}_t)]^{-1}\mathbf{g}(\mathbf{w}_t)$. Furthermore, in order to improve upon the overall efficiency of our SSN algorithms, we will allow for the linear system in the sub-problem to be solved inexactly, i.e., using only a few iterations of any iterative solver such as Conjugate Gradient (CG). Such inexact updates used in many second-order optimization algorithms have been well studied in [4, 5].

As we shall see (in Section 4), our algorithms converge much faster than other competing methods for a variety of problems. In particular, on several machine learning datasets, our methods are at least twice as fast as Newton's methods in finding a high-precision solution while other methods converge slowly. Indeed, this phenomenon is well supported by our theoretical findings—the complexity of our algorithms has a lower dependence on the problem condition number and is immune to any non-uniformity among $\{\mathbf{A}_i(\mathbf{w})\}_{i=1}^{n}$ which may cause a factor of $n$ in the complexity (Table 1). In the following we present details of our main contributions and connections to other prior work. Readers interested in more details should see the technical report version of this conference paper [23] for proofs of our main results, additional theoretical results, as well as a more detailed empirical evaluation.

## 1.1 Contributions and related work

Recently, within the context of randomized second order methods, many algorithms have been proposed that aim at reducing the computational costs involving pure Newton's method. Among them, algorithms that employ uniform sub-sampling constitute a popular line of work [4, 7, 16, 22]. In particular, [19, 20] consider a more general class of problems and, under a variety of conditions, thoroughly study the local and global convergence properties of sub-sampled Newton methods where the gradient and/or the Hessian are uniformly sub-sampled. Our work here, however, is more closely related to a recent work [18](Newton Sketch), which considers a similar class of problems and proposes sketching the Hessian using random sub-Gaussian matrices or randomized orthonormal systems. Furthermore, [1] proposes a stochastic algorithm (LiSSA) that, for solving the sub-problems, employs some unbiased estimators of the inverse of the Hessian.

In light of these prior works, our contributions can be summarized as follows.

- For the class of problems considered here, unlike the uniform sampling used in [4, 7, 19, 20], we employ two non-uniform sampling schemes based on *block norm squares* and a new, and more general, notion of leverage scores named *block partial leverage scores* (Definition 1). It can be shown that in the case of extreme non-uniformity among $\{\mathbf{A}_i(\mathbf{w})\}_{i=1}^{n}$, uniform sampling might require $\Omega(n)$ samples to capture the Hessian information appropriately. However, we show that our non-uniform sampling schemes result in sample sizes completely *independent of $n$* and immune to such non-uniformity.
- Within the context of globally convergent randomized second order algorithms, [4, 20] incorporate inexact updates where the sub-problems are solved only approximately. We extend the study of inexactness to our local convergence analysis.
- We provide a general structural result (Lemma 2) showing that, as in [7, 18, 19], our main algorithm exhibits a linear-quadratic *solution* error recursion. However, we show that by using our non-uniform sampling strategies, the factors appearing in such error recursion enjoy a much better dependence on problem specific quantities, e.g., such as the condition number (Table 2). For example, using block partial leverage score sampling, the factor for the linear term of the error recursion (5) is of order $\mathcal{O}(\sqrt{\kappa})$ as opposed to $\mathcal{O}(\kappa)$ for uniform sampling.
- We demonstrate that to achieve a locally *problem independent* linear convergence rate, i.e., $\|\mathbf{w}_{t+1} - \mathbf{w}^*\| \leq \rho\|\mathbf{w}_t - \mathbf{w}^*\|$ for some fixed $\rho < 1$, our algorithms achieve a lower per-iteration complexity compared to [1, 18, 20] (Table 1). In particular, unlike Newton Sketch [18], which employs random

Table 1: Complexity per iteration of different methods to obtain a problem independent local linear convergence rate. The quantities $\kappa$, $\hat{\kappa}$, and $\bar{\kappa}$ are the local condition numbers, defined in (6), satisfying $\kappa \leq \hat{\kappa} \leq \bar{\kappa}$, at the optimum $\mathbf{w}^*$. $\mathbf{A}$ is defined in Assumption A3 and $\mathbf{sr}(\mathbf{A})$ is the stable rank of $\mathbf{A}$ satisfying $\mathbf{sr}(\mathbf{A}) \leq d$. Here we assume $k_i = 1$, $\mathcal{C} = \mathbb{R}^d$, $R(\mathbf{w}) = 0$, and CG is used for solving sub-problems in our algorithms.

| NAME | COMPLEXITY PER ITERATION | REFERENCE |
|---|---|---|
| Newton-CG method | $\tilde{\mathcal{O}}(\mathrm{nnz}(\mathbf{A})\sqrt{\kappa})$ | [17] |
| SSN (leverage scores) | $\tilde{\mathcal{O}}(\mathrm{nnz}(\mathbf{A})\log n + d^2\kappa^{3/2})$ | **This paper** |
| SSN (row norm squares) | $\tilde{\mathcal{O}}(\mathrm{nnz}(\mathbf{A}) + \mathbf{sr}(\mathbf{A})d\kappa^{5/2})$ | **This paper** |
| Newton Sketch (SRHT) | $\tilde{\mathcal{O}}(nd(\log n)^4 + d^2(\log n)^4\kappa^{3/2})$ | [18] |
| SSN (uniform) | $\tilde{\mathcal{O}}(\mathrm{nnz}(\mathbf{A}) + d\hat{\kappa}\kappa^{3/2})$ | [20] |
| LiSSA | $\tilde{\mathcal{O}}(\mathrm{nnz}(\mathbf{A}) + d\hat{\kappa}\bar{\kappa}^2)$ | [1] |

projections and fails to preserve the sparsity of $\{\mathbf{A}_i(\mathbf{w})\}_{i=1}^n$, our methods indeed take advantage of such sparsity. Also, in the presence of high non-uniformity among $\{\mathbf{A}_i(\mathbf{w})\}_{i=1}^n$, factors $\bar{\kappa}$ and $\hat{\kappa}$ (see Definition (6)) which appear in SSN (uniform) [19], and LiSSA [1], can potentially be as large as $\Omega(n\kappa)$; see Section 3.5 for detailed discussions.

- We numerically demonstrate the effectiveness and robustness of our algorithms in recovering the minimizer of ridge logistic regression on several real datasets (Figures 1 and 2). In particular, our algorithms are at least twice as fast as Newton's methods in finding a high-precision solution while other methods converge slowly.

## 1.2 Notation and assumptions

Given a function $F$, the gradient, the exact Hessian and the approximate Hessian are denoted by $\mathbf{g}$, $\mathbf{H}$, and $\widetilde{\mathbf{H}}$, respectively. Iteration counter is denoted by subscript, e.g., $\mathbf{w}_t$. Unless stated specifically, $\|\cdot\|$ denotes the Euclidean norm for vectors and spectral norm for matrices. Frobenius norm of matrices is written as $\|\cdot\|_F$. By a matrix $\mathbf{A}$ having $n$ blocks, we mean that $\mathbf{A}$ has a block structure and can be viewed as $\mathbf{A} = \left(\mathbf{A}_1^T \cdots \mathbf{A}_n^T\right)^T$, for appropriate size blocks $\mathbf{A}_i$. The tangent cone of constraint set $\mathcal{C}$ at the optimum $\mathbf{w}^*$ is denoted by $\mathcal{K}$ and defined as $\mathcal{K} = \{\Delta | \mathbf{w}^* + t\Delta \in \mathcal{C} \text{ for some } t > 0\}$. Given a symmetric matrix $\mathbf{A}$, the $\mathcal{K}$-restricted minimum and maximum eigenvalues of $A$ are defined, respectively, as $\lambda_{\min}^{\mathcal{K}}(\mathbf{A}) = \min_{\mathbf{x} \in \mathcal{K}\setminus\{\mathbf{0}\}} \mathbf{x}^T\mathbf{A}\mathbf{x}/\mathbf{x}^T\mathbf{x}$ and $\lambda_{\max}^{\mathcal{K}}(\mathbf{A}) = \max_{\mathbf{x} \in \mathcal{K}\setminus\{\mathbf{0}\}} \mathbf{x}^T\mathbf{A}\mathbf{x}/\mathbf{x}^T\mathbf{x}$. The stable rank of a matrix $\mathbf{A}$ is defined as $\mathbf{sr}(\mathbf{A}) = \|\mathbf{A}\|_F^2/\|\mathbf{A}\|_2^2$. We use $\mathrm{nnz}(\mathbf{A})$ to denote number of non-zero elements in $\mathbf{A}$.

Throughout the paper, we make use of the following assumptions:

**A.1 Lipschitz Continuity**: $F(\mathbf{w})$ is convex and twice differentiable with $L$-Lipschitz Hessian, i.e., $\|\mathbf{H}(\mathbf{u}) - \mathbf{H}(\mathbf{v})\| \leq L\|\mathbf{u} - \mathbf{v}\|$, $\quad \forall \mathbf{u}, \mathbf{v} \in \mathcal{C}$.

**A.2 Local Regularity**: $F(\mathbf{x})$ is locally strongly convex and smooth, i.e., $\mu = \lambda_{\min}^{\mathcal{K}}(\mathbf{H}(\mathbf{w}^*)) > 0$, $\quad \nu = \lambda_{\max}^{\mathcal{K}}(\mathbf{H}(\mathbf{w}^*)) < \infty$. Here we define the local condition number of the problem as $\kappa := \nu/\mu$.

**A.3 Hessian Decomposition**: For each $f_i(\mathbf{w})$ in (1), define $\nabla^2 f_i(\mathbf{w}) := \mathbf{H}_i(\mathbf{w}) := \mathbf{A}_i^T(\mathbf{w})\mathbf{A}_i(\mathbf{w})$. For simplicity, we assume $k_1 = \cdots = k_n = k$ and $k$ is independent of $d$. Furthermore, we assume that given $\mathbf{w}$, computing $\mathbf{A}_i(\mathbf{w})$, $\mathbf{H}_i(\mathbf{w})$, and $\mathbf{g}(\mathbf{w})$ takes $\mathcal{O}(d)$, $\mathcal{O}(d^2)$, and $\mathcal{O}(\mathrm{nnz}(\mathbf{A}))$ time, respectively. We call the matrix $\mathbf{A}(\mathbf{w}) = \left(\mathbf{A}_1^T, \ldots, \mathbf{A}_n^T\right)^T \in \mathbb{R}^{nk \times d}$ the augmented matrix of $\{\mathbf{A}_i(\mathbf{w})\}$. Note that $\mathbf{H}(\mathbf{w}) = \mathbf{A}(\mathbf{w})^T\mathbf{A}(\mathbf{w}) + \mathbf{Q}(\mathbf{w})$.

## 2 Main Algorithm: SSN with Non-uniform Sampling

Our proposed SSN method with non-uniform sampling is given in Algorithm 1. The core of our algorithm is based on choosing a sampling scheme $\mathcal{S}$ that, at every iteration, constructs a non-uniform sampling distribution $\{p_i\}_{i=1}^n$ over $\{\mathbf{A}_i(\mathbf{w}_t)\}_{i=1}^n$ and then samples from $\{\mathbf{A}_i(\mathbf{w}_t)\}_{i=1}^n$ to form the approximate Hessian, $\widetilde{\mathbf{H}}(\mathbf{w}_t)$. The sampling sizes $s$ needed for different sampling distributions will be discussed in Section 3.2. Since $\mathbf{H}(\mathbf{w}) = \sum_{i=1}^n \mathbf{A}_i^T(\mathbf{w})\mathbf{A}_i(\mathbf{w}) + \mathbf{Q}(\mathbf{w})$, the Hessian approximation essentially boils down to a matrix approximation problem. Here, we generalize the two popular non-uniform sampling strategies, i.e., leverage score sampling and row norm squares sampling, which are commonly used in the field of randomized linear algebra, particularly for matrix approximation

problems [10, 15]. With an approximate Hessian constructed via non-uniform sampling, we may choose an appropriate solver $\mathcal{A}$ to the solve the sub-problem in Step 11 of Algorithm 1. Below we elaborate on the construction of the two non-uniform sampling schemes.

**Block Norm Squares Sampling**  This is done by constructing a sampling distribution based on the Frobenius norm of the blocks $\mathbf{A}_i$, i.e., $p_i = \|\mathbf{A}_i\|_F^2/\|\mathbf{A}\|_F^2, i = 1, \ldots, n$. This is an extension to the row norm squares sampling in which the intuition is to capture the importance of the blocks based on the "magnitudes" of the sub-Hessians [10].

**Block Partial Leverage Scores Sampling**  Recall standard leverage scores of a matrix $\mathbf{A}$ are defined as diagonal elements of the "hat" matrix $\mathbf{A}(\mathbf{A}^T\mathbf{A})^{-1}\mathbf{A}^T$ [15] which prove to be very useful in matrix approximation algorithms. However, in contrast to the standard case, there are two major differences in our task. First, blocks, not rows, are being sampled. Second, an additional matrix $\mathbf{Q}$ is involved in the target matrix, i.e., $\mathbf{H}$. In light of this, we introduce a new and more general notion of leverage scores, called *block partial leverage scores*.

**Definition 1** (Block Partial Leverage Scores). *Given a matrix* $\mathbf{A} \in \mathbb{R}^{kn \times d}$ *viewed as having* $n$ *blocks of size* $k \times d$ *and a SPSD matrix* $\mathbf{Q} \in \mathbb{R}^{d \times d}$, *let* $\{\tau_i\}_{i=1}^{kn+d}$ *be the (standard) leverage scores of the augmented matrix* $\begin{pmatrix} \mathbf{A} \\ \mathbf{Q}^{\frac{1}{2}} \end{pmatrix}$. *The block partial leverage score for the* $i$-*th block is defined as* $\tau_i^{\mathbf{Q}}(\mathbf{A}) = \sum_{j=k(i-1)+1}^{ki} \tau_j$.

Note that for $k = 1$ and $\mathbf{Q} = \mathbf{0}$, the block partial leverage score is simply the standard leverage score. The sampling distribution is defined as $p_i = \tau_i^{\mathbf{Q}}(\mathbf{A})/\left(\sum_{j=1}^n \tau_j^{\mathbf{Q}}(\mathbf{A})\right), \quad i = 1, \ldots, n$.

---

**Algorithm 1** Sub-sampled Newton method with Non-uniform Sampling

---

1: **Input:** Initialization point $\mathbf{w}_0$, number of iteration $T$, sampling scheme $\mathcal{S}$ and solver $\mathcal{A}$.
2: **Output:** $\mathbf{w}_T$
3: **for** $t = 0, \ldots, T-1$ **do**
4:      Construct the non-uniform sampling distribution $\{p_i\}_{i=1}^n$ as described in Section 2.
5:      **for** $i = 1, \ldots, n$ **do**
6:         $q_i = \min\{s \cdot p_i, 1\}$, where $s$ is the sampling size.
7:         $\widetilde{\mathbf{A}}_i(\mathbf{w}_t) = \begin{cases} \mathbf{A}_i(\mathbf{w}_t)/\sqrt{q_i}, & \text{with probability } q_i, \\ \mathbf{0}, & \text{with probability } 1 - q_i. \end{cases}$
8:      **end for**
9:      $\widetilde{\mathbf{H}}(\mathbf{w}_t) = \sum_{i=1}^n \widetilde{\mathbf{A}}_i^T(\mathbf{w}_t)\widetilde{\mathbf{A}}_i(\mathbf{w}_t) + \mathbf{Q}(\mathbf{w}_t)$.
10:     Compute $\mathbf{g}(\mathbf{w}_t)$
11:     Use solver $\mathcal{A}$ to solve the sub-problem inexactly

$$\mathbf{w}_{t+1} \approx \arg\min_{\mathbf{w} \in \mathcal{C}}\{\frac{1}{2}\langle(\mathbf{w} - \mathbf{w}_t), \widetilde{\mathbf{H}}(\mathbf{w}_t)(\mathbf{w} - \mathbf{w}_t)\rangle + \langle\mathbf{g}(\mathbf{w}_t), \mathbf{w} - \mathbf{w}_t\rangle\}. \tag{2}$$

12: **end for**
13: **return** $\mathbf{w}_T$.

---

## 3  Theoretical Results

In this section we provide detailed complexity analysis of our algorithm.[1]  Different choices of sampling scheme $\mathcal{S}$ and the sub-problem solver $\mathcal{A}$ lead to different complexities in SSN. More precisely, total complexity is characterized by the following four factors: **(i)** total number of iterations $T$ determined by the convergence rate which is affected by the choice of $\mathcal{S}$ and $\mathcal{A}$; see Lemma 2 in Section 3.1, **(ii)** the time, $t_{grad}$, it takes to compute the full gradient $\mathbf{g}(\mathbf{w}_t)$ (Step 10 in Algorithm 1), **(iii)** the time $t_{const}$, to construct the sampling distribution $\{p_i\}_{i=1}^n$ and sample $s$ terms at each iteration (Steps 4-8 in Algorithm 1), which is determined by $\mathcal{S}$; see Section 3.2 for details, and **(iv)** the time $t_{solve}$ needed to (implicitly) form $\tilde{\mathbf{H}}$ and (inexactly) solve the sub-problem at each iteration (Steps 9 and 11 in Algorithm 1) which is affected by the choices of both $\mathcal{S}$ (manifested in the sampling size $s$) and $\mathcal{A}$ see Section 3.2&3.3 for details. With these, the total complexity can be expressed as

$$T \cdot (t_{grad} + t_{const} + t_{solve}). \tag{3}$$

Below we study these contributing factors. Moreover, the per iteration complexity of our algorithm for achieving a problem independent linear convergence rate is presented in Section 3.4 and comparison to other related work is discussed in Section 3.5.

## 3.1 Local linear-quadratic error recursion

Before diving into details of the complexity analysis, we state a structural lemma that characterizes the local convergence rate of our main algorithm, i.e., Algorithm 1. As discussed earlier, there are two layers of approximation in Algorithm 1, i.e., approximation of the Hessian by sub-sampling and inexactness of solving (2). For the first layer, we require the approximate Hessian to satisfy one of the following two conditions (in Section 3.2 we shall see our construction of approximate Hessian via non-uniform sampling can achieve these conditions with a sampling size independent of $n$).

$$\|\widetilde{\mathbf{H}}(\mathbf{w}_t) - \mathbf{H}(\mathbf{w}_t)\| \leq \epsilon \cdot \|\mathbf{H}(\mathbf{w}_t)\|, \tag{C1}$$

or

$$|\mathbf{x}^T (\widetilde{\mathbf{H}}(\mathbf{w}_t) - \mathbf{H}(\mathbf{w}_t))\mathbf{y}| \leq \epsilon \cdot \sqrt{\mathbf{x}^T \mathbf{H}(\mathbf{w}_t)\mathbf{x}} \cdot \sqrt{\mathbf{y}^T \mathbf{H}(\mathbf{w}_t)\mathbf{y}}, \ \ \forall \mathbf{x}, \mathbf{y} \in \mathcal{K}. \tag{C2}$$

Note that (C1) and (C2) are two commonly seen guarantees for matrix approximation problems. In particular, (C2) is stronger in the sense that the spectral of the approximated matrix $\mathbf{H}(\mathbf{w}_t)$ is well preserved. Below in Lemma 2, we shall see such a stronger condition ensures a better dependence on the condition number in terms of the convergence rate. For the second layer of approximation, we require the solver to produce an $\epsilon_0$-approximate solution $\mathbf{w}_{t+1}$ satisfying

$$\|\mathbf{w}_{t+1} - \mathbf{w}_{t+1}^*\| \leq \epsilon_0 \cdot \|\mathbf{w}_t - \mathbf{w}_{t+1}^*\|, \tag{4}$$

where $\mathbf{w}_{t+1}^*$ is the exact optimal solution to (2). Note that (4) implies an $\epsilon_0$-relative error approximation to the exact update direction, i.e., $\|\mathbf{v} - \mathbf{v}^*\| \leq \epsilon\|\mathbf{v}^*\|$ where $\mathbf{v} = \mathbf{w}_{t+1} - \mathbf{w}_t$, $\mathbf{v}^* = \mathbf{w}_{t+1}^* - \mathbf{w}_t$.

**Lemma 2** (Structural Result). *Let $\epsilon \in (0, 1/2)$ and $\epsilon_0$ be given and $\{\mathbf{w}_t\}_{t=1}^T$ be a sequence generated by (2) which satisfies (4). Also assume that the initial point $\mathbf{w}_0$ satisfies $\|\mathbf{w}_0 - \mathbf{w}^*\| \leq \frac{\mu}{4L}$. Under Assumptions A1 & A2, the solution error satisfies the following recursion*

$$\|\mathbf{w}_{t+1} - \mathbf{w}^*\| \leq (1 + \epsilon_0)C_q \cdot \|\mathbf{w}_t - \mathbf{w}^*\|^2 + (\epsilon_0 + (1 + \epsilon_0)C_l) \cdot \|\mathbf{w}_t - \mathbf{w}^*\|, \tag{5}$$

*where $C_l$ and $C_q$ are specified as below.*

- $C_q = \dfrac{2L}{(1 - 2\epsilon\kappa)\mu}$ *and* $C_l = \dfrac{4\epsilon\kappa}{1 - 2\epsilon\kappa}$, *if condition (C1) is met;*
- $C_q = \dfrac{2L}{(1 - \epsilon)\mu}$ *and* $C_l = \dfrac{3\epsilon\sqrt{\kappa}}{1 - \epsilon}$, *if condition (C2) is met.*

## 3.2 Complexities related to the choice of sampling scheme $\mathcal{S}$

The following lemma gives the complexity of constructing the sampling distributions used in this paper. Here, we adopt the fast approximation algorithm for standard leverage scores, [6], to obtain an efficient approximation to our block partial leverage scores.

**Lemma 3** (Construction Complexity). *Under Assumption 3, it takes $t_{const} = \mathcal{O}(\mathrm{nnz}(\mathbf{A}))$ time to construct a block norm squares sampling distribution, and it takes $t_{const} = \mathcal{O}(\mathrm{nnz}(\mathbf{A}) \log n)$ time to construct, with high probability, a distribution with constant factor approximation to the block partial leverage scores.*

The following theorem indicates that if the blocks of the augmented matrix of $\{\mathbf{A}_i(\mathbf{w})\}$ (see Assumption 3) are sampled based on block norm squares or block partial leverage scores with large enough sampling size, (C1) or (C2) holds, respectively, with high probability.

**Theorem 4** (Sufficient Sample Size). *Given any $\epsilon \in (0, 1)$, the following statements hold:*

(i) *Let $r_i = \|\mathbf{A}_i\|_F^2$, $i = 1, \ldots, n$, set $p_i = r_i / (\sum_{j=1}^n r_j)$ and construct $\widetilde{\mathbf{H}}$ as in Steps 5-9 of Algorithm 1. Then if $s \geq 4\mathbf{sr}(\mathbf{A}) \cdot \log (\min\{4\mathbf{sr}(\mathbf{A}), d\}/\delta) /\epsilon^2$, with probability at least $1 - \delta$, (C1) holds.*

(ii) *Let $\{\hat{\tau}_i^{\mathbf{Q}}(\mathbf{A})\}_{i=1}^n$ be some overestimate of the block partial leverage scores, i.e., $\hat{\tau}_i^{\mathbf{Q}}(\mathbf{A}) \geq \tau_i^{\mathbf{Q}}(\mathbf{A})$, $i = 1, \ldots, n$ and set $p_i = \hat{\tau}_i^{\mathbf{Q}}(\mathbf{A}) / (\sum_{j=1}^n \hat{\tau}_j^{\mathbf{Q}}(\mathbf{A}))$, $i = 1, \ldots, n$. Construct $\widetilde{\mathbf{H}}$ as in Steps 5-9 of Algorithm 1. Then if $s \geq 4 \left( \sum_{i=1}^n \hat{\tau}_i^{\mathbf{Q}}(\mathbf{A}) \right) \cdot \log (4d/\delta) /\epsilon^2$, with probability at least $1 - \delta$, (C2) holds.*

**Remarks:** Part (i) of Theorem 4 is an extension of [10] to our particular augmented matrix setting. Also, as for the exact block partial leverage scores we have $\sum_{i=1}^{n} \tau_i^{\mathbf{Q}}(\mathbf{A}) \leq d$, part (ii) of Theorem 4 implies that, using exact scores, less than $\mathcal{O}(d \log d / \epsilon^2)$ blocks are needed for (**C2**) to hold.

### 3.3 Complexities related to the choice of solver $\mathcal{A}$

We now discuss how $t_{solve}$ in (3) is affected by the choice of the solver $\mathcal{A}$ in Algorithm 1. The approximate Hessian $\widetilde{\mathbf{H}}(\mathbf{w}_t)$ is of the form $\widetilde{\mathbf{A}}^T \widetilde{\mathbf{A}} + \mathbf{Q}$ where $\widetilde{\mathbf{A}} \in \mathbb{R}^{sk \times d}$. As a result, the complexity for solving the sub-problem (2) essentially depends on the choice $\mathcal{A}$, the constraint set $\mathcal{C}$, $s$ and $d$, i.e., $t_{solve} = \mathcal{T}(\mathcal{A}, \mathcal{C}, s, d)$. For example, when the problem is unconstrained ($\mathcal{C} = \mathbb{R}^d$), CG takes $t_{solve} = \mathcal{O}(sd\sqrt{\kappa_t} \log(1/\epsilon))$ to return a solution with approximation quality $\epsilon_0 = \sqrt{\kappa_t}\epsilon$ in (4) where $\kappa_t = \lambda_{\max}(\widetilde{\mathbf{H}}(\mathbf{w}_t))/\lambda_{\min}(\widetilde{\mathbf{H}}(\mathbf{w}_t))$.

### 3.4 Total complexity per iteration

Lemma 2 implies that, by choosing appropriate values for $\epsilon$ and $\epsilon_0$, SSN inherits a local constant linear convergence rate, i.e., $\|\mathbf{w}_{t+1} - \mathbf{w}^*\| \leq \rho\|\mathbf{w}_t - \mathbf{w}^*\|$ with $\rho < 1$. The following Corollary gives the total complexity per iteration of Algorithm 1 to obtain a locally linear rate.

**Corollary 5.** *Suppose $\mathcal{C} = \mathbb{R}^d$ and CG is used to solve the sub-problem (2). Then under Assumption 3, to obtain a constant local linear convergence rate with a constant probability, the complexity per iteration of Algorithm 1 using the block partial leverage scores sampling and block norm squares sampling is $\tilde{\mathcal{O}}(\mathrm{nnz}(\mathbf{A}) \log n + d^2 \kappa^{3/2})$ and $\tilde{\mathcal{O}}(\mathrm{nnz}(\mathbf{A}) + \mathbf{sr}(\mathbf{A})d\kappa^{5/2})$, respectively.* [2]

### 3.5 Comparison with existing similar methods

As discussed above, the sampling scheme $\mathcal{S}$ plays a crucial role in the overall complexity of SSN. We first compare our proposed non-uniform sampling schemes with the uniform alternative [20], in terms of complexities $t_{const}$ and $t_{solve}$ as well as the quality of the locally linear-quadratic error recursion (5), measured by $C_q$ and $C_l$. Table 2 gives a summary of such comparison where, for simplicity, we assume that $k = 1$, $\mathcal{C} = \mathbb{R}^d$, and a direct solver is used for the linear system sub-problem (2). Also, throughout this subsection, for randomized algorithms, we choose parameters such that the failure probability is a constant. One advantage of uniform sampling is its simplicity of construction. However, as shown in Section 3.2, it takes nearly input-sparsity time to construct the proposed non-uniform sampling distribution. In addition, when rows of $\mathbf{A}$ are very non-uniform, i.e., $\max_i \|\mathbf{A}_i\| \approxeq \|\mathbf{A}\|$, uniform scheme requires $\Omega(n)$ samples to achieve (**C1**). It can also be seen that for a given $\epsilon$, row norm squares sampling requires the smallest sampling size, yielding the smallest $t_{solve}$ in Table 2. More importantly, although either (**C1**) or (**C2**) is sufficient to give (5), having (**C2**) as in SSN with leverage score sampling yields constants $C_q$ and $C_l$ with much better dependence on the local condition number, $\kappa$, than other methods. This fact can drastically improve the performance of SSN for ill-conditioned problems; see Figure 1 in Section 4.

Table 2: Comparison between standard Newton's methods and sub-sampled Newton methods (SSN) with different sampling schemes. $C_q$ and $C_l$ are the constants appearing in (5), $\mathbf{A}$ is the augmented matrix of $\{\mathbf{A}_i(\mathbf{w})\}$ with stable rank $\mathbf{sr}(\mathbf{A})$, $\kappa = \nu/\mu$ is the local condition number and $\tilde{\kappa} = L/\mu$. Here, we assume that $k = 1$, $\mathcal{C} = \mathbb{R}^d$, and a direct solver is used in Algorithm 1.

| NAME | $t_{const}$ | $t_{solve} = sd^2$ | $C_q$ | $C_l$ |
|---|---|---|---|---|
| Newton's method | $0$ | $\mathcal{O}(nd^2)$ | $\tilde{\kappa}$ | $0$ |
| SSN (leverage scores) | $\mathcal{O}(\mathrm{nnz}(\mathbf{A}) \log n)$ | $\tilde{\mathcal{O}}((\sum_i \tau_i^{\mathbf{Q}}(\mathbf{A}))d^2/\epsilon^2)$ | $\frac{\tilde{\kappa}}{1-\epsilon}$ | $\frac{\epsilon\sqrt{\kappa}}{1-\epsilon}$ |
| SSN (row norm squares) | $\mathcal{O}(\mathrm{nnz}(\mathbf{A}))$ | $\tilde{\mathcal{O}}(\mathbf{sr}(\mathbf{A})d^2/\epsilon^2)$ | $\frac{\tilde{\kappa}}{1-\epsilon\kappa}$ | $\frac{\epsilon\kappa}{1-\epsilon\kappa}$ |
| SSN (uniform) [20] | $\mathcal{O}(1)$ | $\tilde{\mathcal{O}}\left(nd^2 \frac{\max_i \|\mathbf{A}_i\|^2}{\|\mathbf{A}\|^2}/\epsilon^2\right)$ | $\frac{\tilde{\kappa}}{1-\epsilon\kappa}$ | $\frac{\epsilon\kappa}{1-\epsilon\kappa}$ |

Next, recall that in Table 1, we summarize the per-iteration complexity needed by our algorithm and other similar methods [20, 1, 18] to achieve a given local linear convergence rate. Here we provide more details. First, the definition of various notions of condition number used in Table 1 is given below. For any given $\mathbf{w} \in \mathbb{R}^d$, define

$$\kappa(\mathbf{w}) = \frac{\lambda_{\max}(\sum_{i=1}^{n} \mathbf{H}_i(\mathbf{w}))}{\lambda_{\min}(\sum_{i=1}^{n} \mathbf{H}_i(\mathbf{w}))}, \hat{\kappa}(\mathbf{w}) = n \cdot \frac{\max_i \lambda_{\max}(\mathbf{H}_i(\mathbf{w}))}{\lambda_{\min}(\sum_{i=1}^{n} \mathbf{H}_i(\mathbf{w}))}, \bar{\kappa}(\mathbf{w}) = \frac{\max_i \lambda_{\max}(\mathbf{H}_i(\mathbf{w}))}{\min_i \lambda_{\min}(\mathbf{H}_i(\mathbf{w}))}, \quad (6)$$

assuming that the denominators are non-zero. It is easy to see that $\kappa(\mathbf{w}) \leq \hat{\kappa}(\mathbf{w}) \leq \bar{\kappa}(\mathbf{w})$. However, the degree of the discrepancy among these inequalities depends on the properties of $\mathbf{H}_i(\mathbf{w})$. Roughly speaking, when all $\mathbf{H}_i(\mathbf{w})$'s are "similar", one has that $\lambda^{\mathcal{K}}_{\max}(\sum_{i=1}^{n} \mathbf{H}_i(\mathbf{w})) \approx \sum_{i=1}^{n} \lambda^{\mathcal{K}}_{\max}(\mathbf{H}_i(\mathbf{w})) \approx n \cdot \max_i \lambda^{\mathcal{K}}_{\max}(\mathbf{H}_i(\mathbf{w}))$, and thus $\kappa(\mathbf{w}) \approx \hat{\kappa}(\mathbf{w}) \approx \bar{\kappa}(\mathbf{w})$. However, in many real applications, such uniformity doesn't simply exist. For example, it is not hard to design a matrix $\mathbf{A}$ with non-uniform rows such that for $\mathbf{H} = \mathbf{A}^T\mathbf{A}$, $\hat{\kappa}$ and $\bar{\kappa}$ are larger than $\kappa$ by a factor of $n$. This implies although SSN with leverage score sampling has a quadratic dependence on $d$, its dependence on the condition number is significantly better than all other methods such as SSN (uniform) and LiSSA. Moreover compared to Newton's method, all these stochastic variants replace the coefficient of the leading term, i.e., $\mathcal{O}(nd)$, with some lower order terms that only depend on $d$ and condition numbers (assuming $\mathrm{nnz}(\mathbf{A}) \approx nd$). Therefore, one should expect these algorithms to perform well when $n \gg d$ and the problem is moderately conditioned.

## 4 Numerical Experiments

We consider an estimation problem in GLMs with Gaussian prior. Assume $\mathbf{X} \in \mathbb{R}^{n \times d}, \mathbf{Y} \in \mathcal{Y}^n$ are the data matrix and response vector. The problem of minimizing the negative log-likelihood with ridge penalty can be written as

$$\min_{\mathbf{w} \in \mathbb{R}^d} \sum_{i=1}^{n} \psi(\mathbf{x}_i^T\mathbf{w}, y_i) + \lambda\|\mathbf{w}\|_2^2,$$

where $\psi : \mathbb{R} \times \mathcal{Y} \to \mathbb{R}$ is a convex cumulant generating function and $\lambda \geq 0$ is the ridge penalty parameter. In this case, the Hessian is $\mathbf{H}(\mathbf{w}) = \sum_{i=1}^{n} \psi''(\mathbf{x}_i^T\mathbf{w}, y_i)\mathbf{x}_i\mathbf{x}_i^T + \lambda\mathbf{I} := \mathbf{X}^T\mathbf{D}^2(\mathbf{w})\mathbf{X} + \lambda\mathbf{I}$, where $\mathbf{x}_i$ is $i$-th column of $\mathbf{X}^T$ and $\mathbf{D}(\mathbf{w})$ is a diagonal matrix with the diagonal $[\mathbf{D}(\mathbf{w})]_{ii} = \sqrt{\psi''(\mathbf{x}_i^T\mathbf{w}, y_i)}$. The augmented matrix of $\{\mathbf{A}_i(\mathbf{w})\}$ can be written as $\mathbf{A}(\mathbf{w}) = \mathbf{DX} \in \mathbb{R}^{n \times d}$ where $\mathbf{A}_i(\mathbf{w}) = [\mathbf{D}(\mathbf{w})]_{ii}\mathbf{x}_i^T$.

For our numerical simulations, we consider a very popular instance of GLMs, namely, logistic regression, where $\psi(u, y) = \log(1 + \exp(-uy))$ and $\mathcal{Y} = \{\pm 1\}$. Table 3 summarizes the datasets used in our experiments.

Table 3: Datasets used in ridge logistic regression. In the above, $\kappa$ and $\bar{\kappa}$ are the local condition numbers of ridge logistic regression problem with $\lambda = 0.01$ as defined in (6).

| DATASET | CT slices[9] | Forest[2] | Adult[13] | Buzz[11] |
|---|---|---|---|---|
| $n$ | 53,500 | 581,012 | 32,561 | 59,535 |
| $d$ | 385 | 55 | 123 | 78 |
| $\kappa$ | 368 | 221 | 182 | 37 |
| $\hat{\kappa}$ | 47,078 | 322,370 | 69,359 | 384,580 |

We compare the performance of the following five algorithms: (i) *Newton*: the standard Newton's method, (ii) *Uniform*: SSN with uniform sampling, (iii) *PLevSS*: SSN with partial leverage scores sampling, (iv) *RNormSS*: SSN with block (row) norm squares sampling, and (v) *LBFGS-k* is the standard L-BFGS method [14] with history size $k$.

All algorithms are initialized with a zero vector.[3] We also use CG to solve the sub-problem approximately to within $10^{-6}$ relative residue error. In order to compute the relative error $\|\mathbf{w}_t - \mathbf{w}^*\|/\|\mathbf{w}^*\|$, an estimate of $\mathbf{w}^*$ is obtained by running the standard Newton's method for sufficiently long time. Note here, in SSN with partial leverage score sampling, we recompute the leverage scores every 10 iterations. Roughly speaking, these "stale" leverage scores can be viewed as approximate leverage scores for the current iteration with approximation quality that can be upper bounded by the change of the Hessian and such quantity is often small in practice. So reusing the leverage scores allows us to further drive down the running time.

We first investigate the effect of the condition number, controlled by varying $\lambda$, on the performance of different methods, and the results are depicted in Figure 1. It can be seen that in well-conditioned cases, all sampling schemes work equally well. However, as the condition number worsens, the performance of uniform sampling deteriorates, while non-uniform sampling, in particular leverage score sampling, shows a great degree of robustness to such ill-conditioning effect. The experiments shown in Figure 1 are consistent with the theoretical results of Table 2, showing that the theory presented here can indeed be a reliable guide to practice.

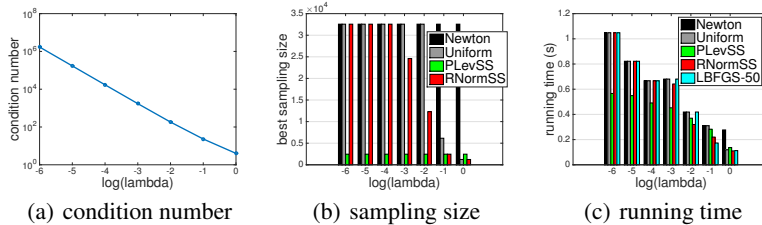

(a) condition number       (b) sampling size       (c) running time

Figure 1: Ridge logistic regression on `Adult` with different $\lambda$'s: (a) local condition number $\kappa$, (b) sample size for different SSN methods giving the best overall running time, (c) running time for different methods to achieve $10^{-8}$ relative error.

Next, we compare the performance of various methods as measured by relative-error of the solution vs. running time and the results are shown in Figure 2[4]. It can be seen that, in most cases, SSN with non-uniform sampling schemes outperforms the other algorithms, especially Newton's method. In particular, uniform sampling scheme performs poorly, e.g., in Figure 2(b), when the problem exhibits a high non-uniformity among data points which is reflected in the difference between $\kappa$ and $\bar{\kappa}$ shown in Table 3.

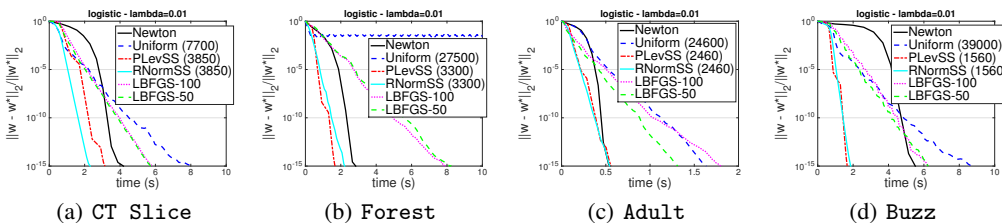

(a) `CT Slice`      (b) `Forest`      (c) `Adult`      (d) `Buzz`

Figure 2: Iterate relative solution error vs. time(s) for various methods on four datasets with ridge penalty parameter $\lambda = 0.01$. The values in brackets denote the sample size used for each method.

We would like to remind the reader that for the locally strongly convex problems that we consider here, one can provably show that the behavior of the error in the loss function, i.e., $F(\mathbf{w}_k) - F(\mathbf{w}^*)/|F(\mathbf{w}^*)|$ follows the same pattern as that of the solution error, i.e., $\|\mathbf{w}_k - \mathbf{w}^*\|/\|\mathbf{w}^*\|$; see [23] for details. As a result, our algorithms remain to be effective for cases where the primary goal is to reduce the loss (as opposed to the solution error).

## 5 Conclusions

In this paper, we propose non-uniformly sub-sampled Newton methods with inexact update for a class of constrained problems. We show that our algorithms have a better dependence on the condition number and enjoy a lower per-iteration complexity, compared to other similar existing methods. Theoretical advantages are numerically demonstrated.

**Acknowledgments.** We would like to thank the Army Research Office and the Defense Advanced Research Projects Agency as well as Intel, Toshiba and the Moore Foundation for support along with DARPA through MEMEX (FA8750-14-2-0240), SIMPLEX (N66001-15-C-4043), and XDATA (FA8750-12-2-0335) programs, and the Office of Naval Research (N000141410102, N000141210041 and N000141310129). Any opinions, findings, and conclusions or recommendations expressed in this material are those of the authors and do not necessarily reflect the views of DARPA, ONR, or the U.S. government.

## Footnotes

[1]In this work, we only focus on local convergence guarantees for Algorithm 1. To ensure global convergence, one can incorporate an existing globally convergent method, e.g. [20], as initial phase and switch to Algorithm 1 once the iterate is "close enough" to the optimum; see Lemma 2.

[2] In this paper, $\tilde{\mathcal{O}}(\cdot)$ hides logarithmic factors of $d$, $\kappa$ and $1/\delta$.

[3]Theoretically, the suitable initial point for all the algorithms is the one with which the standard Newton's method converges with a unit stepsize. Here, $\mathbf{w}_0 = \mathbf{0}$ happens to be one such good starting point.

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
