[Reviews · NeurIPS 2016]

Reviewer 1

Summary

This paper extends the sub-sampled Newton method for objective functions with a finite sum structured. This method possesses comparable convergence rate to Newton's method, yet has much smaller per-iteration cost as they sample functions from the objective. The authors propose to modify this approach by exploiting non-uniform sampling instead of the uniform sampling used in previous papers. The authors analyze two different sampling schemes based on partial leverage scores (commonly used in statistics) as well as the Frobenius norm of a block. The approximate Hessian constructed via non-uniform sampling is used to do an update step which can be solved exactly (this step for example reduces to linear regression in the case of unconstrained problems) or approximately. The bounds derived in the paper show that their sampling method enjoys an improved dependency on the condition number.

Qualitative Assessment

Technical quality: The paper is clearly written and provide a solid theoretical analysis. I just have one minor concern as I could not re-derive Eq. 55 in the appendix. More specifically, I would like the authors to explain how the right-most term appears. Could the authors also comment on whether or not Lemma 2 would reduce to the results derived in [6] in case of uniform sampling? Table 2, last row: Can you explain how you derive the result for t_solve for SSN uniform? Is this counting the construction of Q^t (as defined in [6]) using truncated SVD? Novelty/originality: The convergence results derived in the paper clearly show some improvements for optimization problems that are not well-conditioned. I think it would also be beneficial to contrast the results obtained here to the first-order analogues, e.g. Zhang et al. "Stochastic optimization with importance sampling for regularized loss minimization." (2014). What could be expected of a method that performs non-uniform sampling for both the gradients and the Hessian? Could the authors also comment on what are the practical advantages of their method over using a preconditioner with uniform sampling (for example using a cheap preconditioner such as a diagonal one)? Clarity: The authors mention that “they do not compare with methods in [1, 6] because we can’t replicate the experimental results in their paper.”. Could you please elaborate? Why is it so?

Confidence in this Review

2-Confident (read it all; understood it all reasonably well)


Reviewer 2

Summary

This paper explores new ways to sub-sample datasets for the purpose of approximating the Hessian in Newton's method. Samplers include leverage scores and also row norm scores. This paper ports ideas from the Nystrom approximation literature over the problem of approximating the Hessian. The method is a competitor to sketched Newton methods (which the author mentions but does not compare to) and also stochastic quasi newton methods.

Qualitative Assessment

Pros: This paper is well written and clear. The authors do a good job analyzing their method from a theoretical standpoint. I like that this paper has good theory. I like the kinds of experiments the authors chose, and how they are presented. All in all I think this paper is good, and is a solid contribution to the literature on approximate Newton methods. I have provided some constructive criticism below. Minor comments: 1) The authors don't really have a convergence proof for the method. They show that the method converges under certain assumptions on the Hessian approximation, and that these assumptions hold with high probability on each iteration. However, because the assumptions can fail on some iterations this leaves to door open for non-convergence. It would be better to have a result showing the method converges in expected value, which would be more standard for a non-deterministic optimization algorithm. However, I appreciate that such a result may be hard to prove for this type of algorithm. It may be possible though, given that the parameter lambda gives you a nice lower bound on the singular values that are computed on each iteration. This prevents the singular values of the estimated Hessian from getting too out of whack. But still, such a proof is easier said than done. 2) The method does not provide a big speedup over Newton's method or L-BFGS. Speedup over L-BFGS is about a factor of 2. The authors do not report the number of history updates (k) used for L-BFGS. I anticipate the performance of L-BFGS is sensitive to this choice. Also, given how expensive the Hessian is in this case, it's a good idea to compare to the Composite (i.e., Mehrotra type) Newton method, as this simple modification of Newton's method yields better runtimes when Hessians cost a lot. Major comment: (3) All experiments had very low dimensionality (d < 400). It would be interesting to see how these methods stack up in the higher dimensionality regime. In the rebuttal: I would like the authors to comments on the choice of k they made for L-BFGS and why they made this choice. I'd also like the authors to say something about the limits of the method as d gets large. The authors may choose to respond to minor comment (1), but I'm not expecting a response.

Confidence in this Review

2-Confident (read it all; understood it all reasonably well)


Reviewer 3

Summary

This paper study the problem min sum f_i(x) + q(x) where the hessian of f_i at x is of A_i(x)^t A_i(x) and q(x) is some simple function. They assume that we can compute A(x) efficiently. Let F(x) = sum f_i(x) + q(x). Note that Hessian of F = sum A_i(x)^t A_i(x) + Q(x). If the # of term in the sum is much more than # of dim, it is known that the sum can be approximated by a sampling. The author used this fact on Newton method and propose how to use non-uniform sampling in Newton method and used it to improve the best performance for problems in this form.

Qualitative Assessment

I didn't give a good score because it seems to me is a straightforward application of matrix sampling techniques on Newton method. I would be much happier if the author shows constant/polylog spectral approximation is enough; i.e. the time per iteration is nnz(A) + d^2 sqrt(kappa). --Lemma 2-- Do you think the sqrt(kappa) in the second case is real? Note that (either uniform or non-uniform sampling) gives unbiased estimator for the hessian. If for some magic, the inverse of the estimator is unbiased again, then the sqrt(kappa) term will disappear in expectation. Anyway, seems not easy... --Lemma 3-- How you can get leverage scores in O(nnz(A) log n)? I checked the paper you cite, I guess you are using Theorem 1? They can do it in O~(nd + d^3) time. As far as I know, the fastest algorithm now takes O~(nnz(A) + d^2.38) for example, http://arxiv.org/abs/1408.5099 Maybe I am missing something? I will be super excited if O(nnz(A) log n) can be done. Anyway, it would not affect the result of your paper. It is also easy to show how to do it in O~(nnz(A) + d^2 sqrt(kappa)) Since the newton step need d^2 sqrt(kappa) time anyway, it would be fine. (to do that, you just need to replace every linear system solving in paper http://arxiv.org/abs/1408.5099 by CG)

Confidence in this Review

2-Confident (read it all; understood it all reasonably well)


Reviewer 4

Summary

This paper presents a novel non-uniform subsample scheme for constructing Newton method to solve empirical minimization problems. The authors provide a new algorithm that has better complexity compared to the uniform one and other existing methods. They also provide a contraction inequality that can lead to a local linear convergence of their scheme. A numerical experiment on GLM is given to illustrate the performance of the new method over existing algorithms.

Qualitative Assessment

This paper indeed provides a novel algorithm for non-uniform subsample Newton method due to my knowledge. The theoretical result is new and has a significant impact in this area. In my opinion, the proof of the local convergence of Newton method is relatively standard and can be seen its similarity in many papers. The main difference between such research papers is how to achieve the bound between the true hessian and the approximate one as reflected in the condition (C1) or (C2). The authors propose to use a non-uniform subsample scheme which has a better guarantee in non-uniform setting, especially when matrix A in GLMs is highly non-uniform. The downside of such a scheme is to compute the leverage scores in order to generate the corresponding distribution p_i. In most cases, this has to be recomputed at each iteration due to the change in the loss function. So, it is still unclear to me that how much this cost will affect the overall computational complexity. The authors indeed provide the complexity estimate, but unfortunately, I don't recognize where this was done.

Confidence in this Review

2-Confident (read it all; understood it all reasonably well)


Reviewer 5

Summary

The paper considers the problem of minimizing a sum of function with a separated regularizer. This setting has attracted a lot of attention in past few years, due to its importance in machine learning applications. First order methods in this setting are well studied, but there is still a lot of room for research using second-order methods. The algorithm studied by the authors is a sub-sampled version of the Newton method - instead of computing the true Hessian, one wishes to estimate it using less examples. This method was already considered before. The novelty of the paper lies in the use of a non-uniform sampling distribution for the selection of examples used in the estimate. This idea is very natural, as it is used both in randomized linear algebra and first-order methods in this setting. The authors consider two different sampling strategies, based on distributions used in RandNLA. Both versions are analyzed and they offer improved theoretical guarantees. The paper concludes by an experiment section, where authors showcase the advantage of using their method for ridge logistic regression.

Qualitative Assessment

The paper is very easy to read. The logical flow of the paper is good, everything is stated and defined properly and in the correct order. The tables are very helpful in the understanding. I am missing a theorem, which would state the (number of iterations) * (iteration cost) needed to reach a given precision. The number of iterations and the iteration cost share a dependence on epsilon, which has to be therefore chosen carefully. As far as I can see, this is not a direct corollary of Lemma 2/Corollary 5. Such a result would provide better understanding with the costs associated. The theory and the results are sound, but their influence on the current state-of-the-art is negligible. The authors show linear convergence rate for the proposed methods. For smooth and strongly convex functions this is standard. The best first order methods for this setup reach complexity sqrt(n*kappa)log(1/epsilon) with iteration cost being the average nonzero entries in A_i. In the light of this result, one iteration of the proposed method is too costly. The biggest advantage of Newton methods is their quadratic convergence rate and better dependence on the condition number. As SSN fails to reach this (even with non-uniform samplings), the theoretical results are rather unpractical. Also, the theory assumes that we are already very close to the optimum. If the quadratic convergence does not kick-in, there is no reason to switch from a first-order method. To make a fair comparison, the proposed method outperforms other second-order methods both practically and empirically. That is a nice result - it is definitely a step in the correct direction. However, it does not make a real difference in closing the gap between first-order and second-order methods for large-scale optimization. The influence is therefore practically marginal. The empirical results are clear and it is easy to understand them. It is important to compare against first-order methods, as they are superior to second-order methods in large-scale problems. However, the choice of first order methods is rather poor - GD and AGD are not usable in this setting. The modern methods as SVRG/SDCA/SAG are the current state of the art - with their accelerated counterparts. I am not convinced that SSN would beat these methods. One should definitely include such experiments. Also, it seems to be rather difficult to tune the sampling size - it is a problem dependent quantity, which has to be chosen carefully. It would be interesting to see the behaviour of the method for different choices of sample sizes. The intuition is, that low s will cause the algorithm to diverge, while high s will result in very costly iterations. Is this correct? To conclude, the paper is well written and it shows some nice results, although the real influence is debatable.

Confidence in this Review

2-Confident (read it all; understood it all reasonably well)


Reviewer 6

Summary

The focus of this paper is improving convergence times for Newton's method by approximating the Hessian during each iteration. The proposed algorithm applies to smooth, strongly convex objectives subject to convex constraints. An important assumption is that the Hessian is a sum of known low-rank matrices and one additional matrix. The main idea is to approximate the Hessian by intelligently subsampling (and properly rescaling) the low rank matrices. The Newton step is then solved approximately. Prior works considered uniform sampling for approximating the Hessian. This paper studies novel non-uniform sampling techniques based on the norm or standard leverage scores of the Hessian components. In the theoretical results section, the paper builds off recent work on fast matrix approximation methods to derive sufficient asymptotic sample sizes for the algorithm to achieve a local linear convergence rate with high probability. An important point about this result is that unlike uniform sampling, the sufficient sample size for the proposed method does not depend on the problem's total number of training examples. The section also clearly compares the time complexities of the proposed and related algorithms. The theory allows the different sampling schemes to be compared based on their dependence on the problem's condition number, which is interesting. The paper only focuses on local convergence guarantees and only very briefly refers to other work regarding what to do in the initial phase of the algorithm. In the empirical section, the paper considers logistic regression problems with l2 regularization. The proposed Hessian sampling approaches are compared to uniform sampling, Newton's method, and LBFGS. The experiments focus on obtaining highly accurate solutions for data sets with roughly 100 features and 100k examples. The proposed methods seem to clearly outperform uniform sampling, while standard Newton is a closer competitor.

Qualitative Assessment

There are some obvious criticisms for this paper. First, the algorithm focuses on finding highly precise solutions to loss minimization problems with many more examples than features. For most real world prediction tasks, it is well-known that highly precise solutions are not necessary. Second, it is not clear from the experiments whether the algorithm significantly improves upon the conventional Newton's method. It seems Newton's method consistently displays the same or faster local convergence speed compared to the proposed methods, and Newton's method is only slower in early stages. This observation is not discussed, and there are not sufficient experiment details to understand why this is the case (for example whether the same subproblem solver is used for both methods, which steps of the algorithms dominate iteration times, what type of backtracking is used). This is especially an issue, since the analysis focuses only on the local convergence phase, and it is not clearly stated what the algorithms do in the initial phase (see footnote 1 in the paper). On a related note, the sampling size for each method is chosen to optimize convergence progress during the initial stage of the algorithm (footnote 5 in the paper), and the uniform sampling method looks competitive in this phase. However, the algorithms are then evaluated based on their local convergence speed, calling into question the fairness of the parameter tuning. Finally, compared to Newton's method, the proposed method has additional parameters to tune (sample size, when to use stale leverage scores), and the data set sizes are quite small. For these reasons, I am not convinced the algorithm will be useful in practice. As a whole, however, the paper has several appealing aspects. Assuming that the conclusions of the empirical section are correct, I was surprised that the somewhat complicated partial leverage scores sampling scheme significantly outperforms uniform sampling. I imagine these subsampling and matrix approximation ideas could be useful in other areas of optimization and machine learning, and this paper could serve as a template for applying this ideas. The theory is also interesting in the sense that the different methods can easily be compared based on their dependence on the number of training examples as well as the problem's condition number. The sections of the paper also complement each other fairly well, as the theory can be used to explain several of the empirical results. Finally, the presentation is for the most part very clear. Some suggestions/questions for the authors to potentially consider are: 1. Think about defining the objective as \sum_i f_i(a_i^T w) + R(w), so that the Hessian structure is more explicit. Also, what is the benefit of defining the A_i's as matrices and not vectors (i.e. k_i >= 1 vs. k_i = 1)? This "block" notion was consistently confusing to me, and it is not clear to me what additional objectives this allows. 2. Also consider removing the constraint from the problem formulation. The constraint does not seem to be important to the paper, and it causes some confusion when stating time complexities of the subproblem solvers. 3. Lemma 2 is presented without much discussion. To make the theory section easier to follow, consider briefly discussing the implications of meeting conditions C1 and C2 immediately after the lemma. 4. The sample size (in)dependence on n is framed as an important point. For me at least, it was not obvious how the sr(A) and \tau(A) quantities in the bounds related to n, if at all. It may be helpful to make this more clear in the text. 5. Consider including empirical results (likely in an appendix) that demonstrate the trade-offs between the time required and approximation error for each way to construct the Hessian. This would be helpful in understanding the convergence vs. time curves. 6. It seems to be that for row norm squares sampling in the logistic regression case, the norms of the feature vectors can be cached to perform the sampling in O(n) time. If this is true, it might be worth mentioning. 7. Consider including empirical results that illustrate the sample size bound's independence on n. 8. Think about removing the last paragraph of Section 4 to add additional commentary elsewhere.

Confidence in this Review

2-Confident (read it all; understood it all reasonably well)